# How Do Children with Autism Spectrum Disorder and Children with Developmental Delays Differ on the Child Behavior Checklist 1.5–5 DSM-Oriented Scales?

**DOI:** 10.3390/children9010111

**Published:** 2022-01-14

**Authors:** Yi-Ling Cheng, Ching-Lin Chu, Chin-Chin Wu

**Affiliations:** 1Department of Psychology, Kaohsiung Medical University, Kaohsiung 807378, Taiwan; yilingcheng@kmu.edu.tw; 2Department of Educational Psychology & Counseling, National Pingtung University, Pingtung 900391, Taiwan; 3Department of Medical Research, Kaohsiung Medical University Hospital, Kaohsiung 807377, Taiwan

**Keywords:** autism, CBCL1.5–5, differential item functioning, measurement invariance

## Abstract

The Child Behavior Checklist 1.5–5 (CBCL 1.5–5) is applied to identify emotional and behavioral problems on children with developmental disabilities (e.g., autism spectrum disorder [ASD] and developmental delays [DD]). To understand whether there are variations between these two groups on CBCL DSM-oriented scales, we took two invariance analyses on 443 children (228 children with ASD). The first analysis used measurement invariance and multiple-group factor analysis on the test structure. The second analysis used item-level analysis, i.e., differential item functioning (DIF), to discover whether group memberships responded differently on some items even though underlying trait levels were the same. It was discovered that, on the test structure, the Anxiety Problems scale did not achieve metric invariance. The other scales achieved metric invariance; DIF analyses further revealed that there were items that functioned differently across subscales. These DIF items were mostly about children’s reactions to the surrounding environment. Our findings provide implications for clinicians to use CBCL DSM-oriented scales on differentiating children with ASD and children with DD. In addition, researchers need to be mindful about how items were responded differently, even though there were no mean differences on the surface.

## 1. Introduction

The applications of the Child Behavior Checklist 1.5–5 (CBCL 1.5–5) are beneficial in identifying individuals with developmental disabilities (e.g., autism spectrum disorder [ASD] or developmental delays [DD]). However, fewer studies explored whether the CBCL Diagnostic and Statistical Manual of Mental Disorders (DSM)-oriented scales can be used to identify the distinctions between developmental disabilities, particularly when symptoms could be similar on the surface in early development. Previous studies have found valuable results on comparisons between individuals with ASD and those with DD on early age [1]. Yet, less is known about whether CBCL DSM-oriented scales are sensitive enough to differentiate between these two groups or whether they measure the same construct across groups. The current study explores this question with two invariance analyses to study if there were tangible differences on CBCL DSM-oriented scales at both the test level and the item level when diagnosing these two groups of young children. 

### 1.1. Factor Structure of CBCL DSM-Oriented Scales with ASD and DD

The CBCL DSM-oriented scales were established by Achenbach and colleagues [2]. Compared to the CBCL syndrome scales (which were developed through psychometric analyses), CBCL DSM-oriented scales were constructed through the efforts of a group of experts. Since then, a few attempts were made to validate its structure, and whether it can be used to differentiate and categorize developmental disabilities. Among these attempts, one specific application was to identify typical children, children with other developmental disabilities, or children with ASD. For example, Chericoni and colleagues [3] investigated the use of DSM-oriented scales with 18-month-old toddlers with suspected ASD diagnosis and typical children. In the follow-up study, they found that early assessment of the DSM-Pervasive Developmental Problems (DSM-PDP) scale was effective in identifying children with ASD. These results were encouraging, but the evidence of the psychometric properties of CBCL DSM-oriented scales with ASD children was actually lacking, meaning whether we had measured the same constructs between groups was in question. A few studies have shown the validity of the factor structure of CBCL DSM-oriented scales (e.g., the PDP scale) across different cultures. For example, Rescorla and colleagues [4] examined the factor invariance of the CBCL DSM-ASD scale (which is an identical scale with the PDP scale but without one item included) across 24 countries. They found that strong measurement invariance (scalar invariance) can be held even with such large and diverse samples. However, the merit of Rescorla and colleagues [4] provided is that the same construct can be measured with the children with ASD across different cultures. Whether CBCL DSM-oriented scales measure the same construct between individuals with ASD and DD remained unknown. It could be helpful to explore this question with similar methodologies (e.g., testing for measurement invariance) but compare different clinical samples. In ASD research, an investigation of test invariance modeling is a common method to identify the group differences in the test structure, particularly when comparing children with ASD and those with other developmental disabilities [5,6,7]. In general, investigation of measurement or test invariance is an approach to examine whether construct is identical between groups. Previous studies have demonstrated that application of this method can be useful in showing the variations among samples. For instance, contrasting the ASD group with several Intelligence Quotient (IQ) levels [5] and with different cultures was a common approach [7]. In addition, several screening tools of ASD have undergone the same examination for the investigation of distinct samples (e.g., individuals with ASD vs. other populations). For instance, in a sample of adults with ASD and typical adults, Murray and colleagues [6] tested measurement invariance approaches of the screening tools, such as the Autism Spectrum Quotient Short Form(AQ-S). A study conducted by Glod et al. [8] also examined the parent version for the Spence Children’s Anxiety Scale: Parent Version (SCAS-P) with a measurement invariance approach.

### 1.2. The Application of Differential Item Functioning with Screening

While psychometrics analysis is commonly used to identify the variation between groups, it can be applied on the test level or the item level. One common approach is measurement invariance, which we illustrated above. Aside from measurement invariance, recently, a different type of psychometric analysis on screening tools on the item level of ASD has gained its popularity. The approach, named differential item functioning (DIF), is applied to check the validity of measurements across many disciplines, particularly on disabilities [9,10,11,12]. DIF analysis focuses on the probability of choosing a particular answer, specifically, how participants might respond to items differently because of the group memberships. Hypothetically, if two people have the same level of traits, they should have a similar probability of responding to the same answers on that item and should not be differed by group membership. However, in reality, items sometimes function differently because of background variables, such as gender, developmental disabilities, or socioeconomic status (SES), even when two people have similar trait levels.

DIF has been proven to be useful even when items are categorical or ordered, such as the item types in CBCL (e.g., the Anxiety Problems scale used in the R software package Lordif, which we will mention later). DIF is a broadly defined term, including both the item response theory (IRT) and non-IRT approaches. To date, several directions are developed to identify the item variation [13]. These methods were applied to the use of ASD screening tools. One direction is using DIF to ensure the quality of the measurement. For example, Mazefsky et al. [14] applied DIF analysis on the Emotion Dysregulation Inventory (EDI) to ensure the selected items did not function differently because of age, gender, IQ, and verbal ability. Another direction is using DIF to identify whether the current measurement functioned differently because of group memberships. For instance, Agelink van Rentergem et al. [15] examined whether items in the Autism Spectrum Quotient (AQ) functioned differently between adults with autism and typical adults. With several DIF methods, the study concluded that negatively phrased items (an example item is “I don’t particularly enjoy reading fiction”) often functioned differently between people with autism and without autism. These two studies used adults as the subjects, but the same methodology was applied to screening tools for young children. In fact, McClain et al. [16] used IRT DIF analysis on Autism Spectrum Rating Scales (ASRS) with a group of children (aged 6–18). They have found that several items functioned differently with ethnic group memberships. Specifically, these items reflected distinctive meanings for parents with different racial backgrounds even the children have similar severity of symptoms. This has made people wonder if DIF can be applied to the screening tools for younger children with developmental disabilities. There are already some attempts. For example, Lazenby et al. [17] used DIF methods and found that 12-month-old infants who were at high risk for ASD showed different language development compared to the non-ASD group. Another study executed by Visser and colleagues [18] also suggested that DIF methods were useful in differentiating children aged 1.5–4 with developmental disabilities and typical children. This demonstrated that DIF can be a practical method to identify developmental differences even with very young children. Overall, this suggested the possibility to apply the same methodology between different developmental disabilities (e.g., the ASD group and the DD group) on screening tools for young children (e.g., CBCL).

### 1.3. The Present Study

In order to fill the gap from previous studies, the present study proposed to use two approaches to explore whether the validity of CBCL DSM-oriented scales can be endorsed for the use of differentiation between ASD children and DD children.

## 2. Materials and Methods

### 2.1. Participants

Participants were recruited from a teaching hospital in Chai-Yi city of Taiwan. At that time, 443 individuals with suspected developmental problems and their parents agreed to join the study. Participants were assessed by an experienced team. The team was with a group of experts (i.e., advanced psychiatrists and clinical psychologists). Information on parents’ current concerns, the test results on cognitive and adaptive functioning, children’s developmental histories, children’s behaviors in the clinical setting, and the findings of the Autism Diagnostic Observation Schedule (ADOS) [19] were evaluated together. Based on joined judgments, children were then designated into two subgroups. The ASD group consisted of 228 (girls = 25) participants. The mean age of this group was around 32.28 months (standard deviation = 9.16). These children were diagnosed based on the DSM-5 criteria [20] and they exhibited a minimum of three deficits in social communication and social interaction and two restricted/repetitive behaviors. In addition, these children also belonged to autism or non-autism ASD based on the ADOS classification. The DD group was 215 (girls = 66) children. First, these participants should not fit ASD criteria in the DSM-5. The diagnostic criteria were from a joined judgment, including under a total score of 85 on the Mullen Scales of Early Learning (MSEL) [21] or a T-score of 35 on any cognitive scales of the MSEL. The mean age of the DD group was around 30.75 months (standard deviation = 10.10).

### 2.2. Procedure and Measures

Children’s parents were asked to fill out the CBCL 1.5–5 [22]. The purpose of CBCL 1.5–5 is to identify a set of behavioral and emotional problems in children. It has been standardized and used all around the world. The Chinese version is a translated version that went through the standardized language clarification procedure from the English version of CBCL. The Chinese version of CBCL had strict psychometric evaluations. For reliability, the Cronbach’s alpha was found above 0.70 in several diverse samples [23]. Using preschool kids in Taiwan as samples, the test–retest reliability of the CBCL 1.5–5 was 0.52–0.84 [23,24]. The Chinese version has the same item format, such as the same 99 items. These were also ordered as how they were in English version. These items are selected and organized into 5 DSM-oriented scales: Affective Problems, Anxiety Problems, PDP, Attention-Deficit/Hyperactive Problems (ADHP), and Oppositional Defiant Problems (ODP).

To give an estimation of mental age of our participants, these children received the test of the MSEL [21]. The MSEL is an assessment battery to measure development of children between birth and 68 months of age. The test has four sets of cognitive scales: Visual reception, fine motor, receptive language, and expressive language. These cognitive scales can be derived with T-scores. These cognitive scale scores could be combined as total scores. This composite score can serve as an indicator of cognitive abilities. To generate an estimation of mental age, all children were computed by averaging the age equivalents of the four cognitive scales.

The ADOS [19] is a play-based, interactive, and semi-structured tool. It has four modules, and each module is decided and executed depending on the age and expressive language of a child. The ADOS is thought to be the best diagnostic test for ASD and provides a standardized opportunity of observing and rating communication as well as reciprocal social interaction which together form the communication social total scores. Each set of the scales provides a way to calculate cutoff scores. Using cutoff scores, three categories (i.e., autism, non-autism ASD, or non-ASD) can be designated to the examinees.

### 2.3. Data Analysis

Descriptive statistics were presented first for readers to provide an idea of the characteristics of participants. We also compute the reliability of each subscale. CBCL 1.5–5 is a standardized measure. The manual has documented the established validity and reliability from previous studies, but the initial use of the CBCL is not designed for the use of participants that suspected developmental-related disabilities (despite the fact that, over the years, the measure was used for this kind of research aim, such as evidence-based practice). Therefore, to give more information about our current sample, we conducted our reliability estimations. This was the consideration we took from the book “*Standards for Educational and Psychological Testing*” [25]. One commonly used reliability estimate, called Cronbach’s alpha, was presented. We also provided additional reliability estimates, such as the greatest lower bound (glb) [26]. Both were calculated, respectively, on each subscale. As for the more favourable standard, alpha is a commonly found procedure in most psychological or social science studies, but glb provides an idea of the interval estimation of true reliability. Particularly, it can be positioned between the value of glb and 1 [27]. JASP 0.14.1 [28] was used for the calculations of these values. 

#### 2.3.1. Measurement Invariance

To examine the similarities and differences of the factor structure among different samples, measurement or test invariance is a common modeling approach that is utilized in many disciplines [29]. In order to identify the sources of variations, a set of factor analyses progress steps by steps on checking the critical features between groups. Therefore, sometimes in previous publications, multiple-group confirmatory factor analysis is the alternative name for this type of test invariance modeling [30].

Specifically, the approach of this invariance analysis investigates the representations of the psychological processes across different samples. Methodologically, three steps are used across studies for measurement invariance: configural invariance, metric invariance, and scalar invariance. At first, a loose model will be conducted, followed by a stricter model, which should continue until the stricter model fit becomes worse than the previous model. Configural invariance identifies whether the similar factor structures can be found between samples. Metric invariance checks when the factor loadings can be identical (set as the same) across samples. Scalar invariance investigates whether the identical means of the ability (which are intercepts in the model) can be found between groups [31]. The major stopping rule is that the comparisons had to stop when the fitting criteria indicated that the following model is significantly inferior to the previous model. For instance, when the comparison showed scalar invariance in a significantly worse fit (as to metric invariance), the result of scalar invariance cannot be proceeded. It is a particularly beneficial method to examine whether there are group differences on a measurement, possibly due to background factors. For the constructs to be validated as “measurement invariant”, considerations or opinions on what is the level of invariances that needs to be accomplished have been diverse with different research questions being asked [29]. While the focus of the comparison might vary across studies, the justification of the levels of invariances should be identified first [32]. For the current study, we want to achieve metric invariance. Our main focus is to understand whether the relationships between each question and psychological constructs are the same between the ASD sample vs. the DD sample.

The analysis used the R Lavaan package [33]. Lavaan is a statistical package for structural equation modeling (SEM). The functions of measurement invariance of Lavaan are comparable with Mplus [34]. Mplus is a commercial statistical package, known as a powerful analytic tool for SEM. Because R software is open source, the validity of the software is important. We found that, until the year 2020, the evaluation of Lavaan showed it is still in excellent condition [35].

Estimation method: To proceed with measurement invariance, the first step decides which estimation method should be applied. This is because items in CBCL scales range from 0 to 2. This resulted in a non-normal distribution. We applied the weighted least squares mean and variance (WLSMV) estimator for our analysis, since it is specifically designed for categorical item responses. Just to iterate the process of decision making, we considered that both the maximum likelihood parameter estimates standard errors (MLR), and that WLSMV with robust estimation can be used for dealing with this issue [36,37]. However, for the purpose and item type of this study, WLSMV is more suitable.

Fit indexes: Two sets of model fit indicators were used to explore the fit of the models. The absolute model fit was used to distinguish the fits of unidimensional model of each subscale as well or not. For this type of analysis, we used two criteria. The value of the Root Mean Square Error of Approximation (RMSEA) is the primary criteria. Comparative fit index (CFI) is the second. RMSEA should be under 0.08 to qualify a modest fit, and under 0.05 can be considered as an excellent fit. The CFI index needs to be over 0.95 to be evaluated as excellent fits [38,39]. The relative model fits were used to compare model fits between measurement invariance models. The chi-square difference test was the first criteria. After that, we checked differences on alternative indices, including CFI, RMSEA, and SRMR (standardized root mean square residual), as a comparison [31,40]. Meade and colleagues [41] found that the group sample size is a critical factor to choose criteria. To be more specific, a significant chi-square difference test with a sample size over 200 did not mean variant models. While other fit indices showed values well under the criteria, this comparison should be considered as a measurement invariance. A justification is that a relatively huge sample size might result in the high possibility of test values on chi-square [42]. The sample sizes in the current study are 228 and 215. These are right on the edge of 200; therefore, we still applied the test values of the chi-square difference test for the primary criteria, and other indexes were also weighted in. Specifically, a combination of a significant chi-square test and one over-the-standards alternative index would lead the model comparison to be evaluated as non-invariance. Second, a combination of non-significant chi-square test with two over-the-standards alternative fit indices (any two indices of CFI, RMSEA, and SRMR) would lead the comparison to be evaluated as non-invariance. In addition to these rules, we also consider how different models might have different criteria. We followed Chen’s [40] judgements. The sample size of the current study was 443 (over 300), and it was the best for us to use the criteria as follows: *p* value (≤0.05) on chi-square and/or the changed value on CFI is ≥−0.010, RMSEA is ≥0.015, and SRMR is ≥0.030. This result of the model comparisons should be considered as worse fits. In addition, for testing scalar (intercept) invariance, CFI is ≥−0.010, RMSEA is ≥0.015, and SRMR is ≥0.010(SRMR is stricter). This result can be considered as worse fits [36]. In addition, a few papers recommended that applying partial invariance can be a possible solution after full invariance could not be achieved [29,43,44]. Partial invariance can be an approach to unwind the possible parameters to fix the issue of model fits. Most software has the function to generate the possible modification index, but we considered that it might not be the best strategy for our study. This is because, sometimes, such an attempt can potentially cause type I errors [44]. Particularly, in our situation, there were not many papers compared to the ASD group and the DD group. We are not sure if the choice to relax particular parameters is a correct move. Therefore, the study was proceeded with only full invariance models.

#### 2.3.2. Differential Item Functioning

To further locate the possible item variations between the ASD sample and the DD sample, we performed DIF analysis. Because items in CBCL are ordinal, we used the R software package, Lordif [13], to conduct an analysis that can deal with the particular characteristics of ordinal data. The Lordif package used a logistic regression with the IRT-a hybrid approach. The procedure of the analysis followed closely with what was carried out in Choi et al.’s example [13]. DIF is a method which shows how items might function differently when the participants’ underlying abilities are at the same levels. There are many methods of DIF. Item response theory (IRT) models and non-item response theory models have both been developed over the years. In DIF, items are also identified as two types: uniform DIF and non-uniform DIF. The difference between these two types of DIF is that the uniform DIF item shows a DIF effect across all levels of abilities, whereas a non-uniform DIF item only shows the DIF effect on a certain level of abilities (e.g., low-ability or high-ability students). 

## 3. Results

### 3.1. Descriptive Statistics

Table 1 presented the demographic comparisons of the ASD group vs. the DD group. Between groups, we found that there were significant differences in some variables. For example, the DD group was better on mental ages. Parents in the ASD group are more educated. The ASD group also had higher ADOS scores, which is expected. One father of children with DD was missing a value on the education variable. On the ratio of gender, the ASD group also has more males (*p* < 0.001). By applying Bonferroni statistical correction (dividing 0.05 by 8 = 0.006), as the analysis used multiple *t*-test comparisons with eight variables, these variables still remained significant. 

### 3.2. Reliability Estimates

The Cronbach’s alphas and glbs of each subscale were described as *Scale (alpha/glb)*: Affective Problems ($0.65_{ASD},\;0.72_{DD}$)/($0.76_{ASD},\;\;0.81_{DD}$), Anxiety Problems $(0.73_{ASD},\;0.68_{DD}$)/($0.82_{ASD},\;0.78_{DD}$), PDP ($0.75_{ASD},\;0.68_{DD}$)/($0.85_{ASD},\;0.82_{DD}$), ADHP ($0.68_{ASD},\;0.69_{DD}$)/($0.80_{ASD},\;0.81_{DD}$), and ODP ($0.80_{ASD},\;0.83_{DD}$)/($0.86_{ASD},\;0.87_{DD}$). The alphas were quite different between the ASD group and the DD group on the first three scales. This suggests that the ratio between variance on items and variation on total scales was dissimilar between these two groups on these three scales. In addition, Affective Problems showed slightly lower alphas (0.65) on children with ASD, suggesting that the consistency of items in this scale might be the lowest among these subscales. We also observed similar patterns with the values of glb.

### 3.3. Measurement Invariance

The complete results of measurement invariance model comparisons are shown on Table 2. The details of factor loadings of each subscale appeared on Appendix A and Table A1. At the first look, only the Anxiety Problems scale did not achieve metric invariance. All other scales achieved metric invariance. However, none of the scales achieved scalar invariance. The results suggested that the item loadings were similar across groups on these subscales but the means of items between groups were different. Based on the results, we concluded that the relations between test items and psychological constructs were identical across children with ASD and those with DD, suggesting that the components of psychological constructs were similar across groups. However, in lieu of none of these scales achieved scalar invariance, this suggested that the use of CBCL to differentiate the ASD sample vs. the DD sample needs to be careful if the purpose is to compare the means of the observed scores as they might not be equivalent at the first place.

### 3.4. Differential Item Functioning

Next, for these subscales that passed the measurement invariance, in order to understand whether there might be differences on items-person responses in each subscale, we conducted DIF analysis. The complete results of DIF on each subscale are shown in Table 3 and Figure 1. As mentioned earlier in the method section, DIF items suggested that the group performed differently even when the underlying traits are at the same level. A simple example could be two students with the same level of severity on anxiety whereby one responded to an item with 1 and another responded to an item with 2 on CBCL. As such, there are a few items flagged in these subscales. Specifically, in Affective Problems, CBCL 49 and CBCL 71 are flagged. This implies that the ASD group and the DD group responded differently on these two items when the levels of the latent traits of Affective Problems are the same. Repeating with the same DIF analysis, we found that, in PDP, CBCL 21 and CBCL 92 were flagged. In ADHP, CBCL 6 and CBCL 36 were flagged. In ODP, CBCL 85 and CBCL 88 were flagged.

## 4. Discussion

The purpose of the study was to examine the clinical use of CBCL on different developmental disabilities (ASD vs. DD). Particularly, we explored whether factor structures of CBCL DSM-oriented scales were similar across the ASD group and the DD group. We also further explored whether there were items functioned differently when underlying traits were at the same level between groups. Overall, our results suggested that, when using CBCL DSM-oriented scales to differentiate between the ASD group and the DD group, it might be helpful to use a subscale level. The Anxiety Problems scale should be used on an item level. In addition, among the subscales that achieved metric invariance, there were particular items responses that were acted differently. We discuss these differences and their implications for clinical use of CBCL on the ASD group and the DD group below.

First, in order to use the whole CBCL DSM-oriented test, we found that the factor structures of all CBCL subtests were similar across these five scales, and that the connections between test questions and psychological constructs were also identical across groups among four scales, except the Anxiety Problems scale. The Anxiety Problems scale was the only scale that did not pass the test of metric invariance. However, the fact that other scales achieved metric invariance suggested a certain level of measurement invariance was accomplished. However, none of the scales passed the test of scalar invariance, which advised that we only achieve weak measurement invariances. Therefore, when comparing the ASD group and the DD group, the use of CBCL DSM-oriented scales can be helpful, but it is with some limitations. 

Our findings on the lack of invariance over the Anxiety Problems scale echoed previous findings on the different anxiety levels between these two clinical groups. Previous studies on the comparison between the ASD group vs. the DD group on CBCL scales have found that children with ASD sometimes could possess high anxiety. A few studies [45] have all suggested that children with ASD have a high level of anxiety; these studies were with both older children (age over 6 years old) and younger children (children under age 6). One study found that children with ASD have higher anxiety compared to children with DD [46].

With the analysis of DIF on the subscales passed metric invariance, we further discovered that several items were flagged as they functioned differently between the ASD group and the DD group. Upon further inspection, we discovered that these items are perhaps related to the different symptoms between groups. The identification of DIF items is decided with the criteria that people who possess the same level of latent traits act differently on these items. Therefore, from this finding, we can speculate that children with ASD and those with DD showed different responses patterns on certain items, even when they have the same estimated latent traits. For example, one group of DIF items showed that they are mostly related to children’s reactions to the environment. Specifically, these items are CBCL 21 (disturbed by any change in routine), CBCL 92 (upset by new people or situations), and CBCL 71 (shows little interest in things around him/her). Intriguingly, though it looks like these items corresponded to the symptoms that children with ASD commonly have but not for children with DD, we found the patterns were in fact not all consistent. These two items in PDP, i.e., CBCL 21 (disturbed by any change in routine) and CBCL 92 (upset by new people or situations), showed that children with DD responded to a higher score category when both groups were in the same latent trait level. Previous studies found mixed results. Rescorla and colleagues [47] found that children with ASD received higher scores on this scale, while Predescu and colleagues [48] found no difference. With a further inspection, we found that, on these two items, children with ASD had higher mean scores (CBCL 21-ASD:.57/DD:.51, CBCL92- ASD:.68/DD:.65); however, from the perspective of DIF analysis with the concept of latent trait that was taken into consideration, it was found that parents of children with DD responded with higher scores on these two items. One explanation might be that, when asked to evaluate their children with the questions in the PDP subscale compared to other items, these two items were relatively salient for the parents of children with DD. In turn, they reflected the responses more intensely on these two questions. Furthermore, on CBCL 71 (shows little interest in things around him/her) in Affective Problems, we found that children with ASD responded with higher scores (see Figure 1B). In addition, there is a second group of items showed that those children with DD mostly may act upon in these situations, compared to children with ASD. For example, items CBCL 6 (cannot sit still, restless, or hyperactive) and CBCL 36 (gets into everything) are two of the items flagged here. For CBCL 6 (cannot sit still, restless, or hyperactive), we found that, although children with DD at high-level trait did respond with higher scores compared to ASD, we also found that opposite patterns children with DD at low-level trait responded with lower scores compared to ASD.. For CBCL 36 (gets into everything), we also found that children with DD responded with higher scores, though the lines were pretty flap here and had no discrimination power. It is possible that the meaning of “everything” was not clear and opened to interpretation. These two sets of items seemed to represent totally opposite behavioral patterns on the surface; however, they both characterize children’s responses to the environment. Thus, they were perhaps exemplified as the unique or sensitive indices that we can utilize to differentiate children with ASD from children with DD. 

In addition, there were two items, i.e., CBCL 85 (temper tantrums or hot temper) and CBCL 88 (uncooperative), in the ODP scale which were flagged as DIF items. However, these two items seemed to show symptoms on both the ASD group and the DD group in previous studies. In fact, in our analysis, we found that, when they received the same level of the latent trait, for CBCL 85 (temper tantrums or hot temper), children with DD responded to higher scores compared to children with ASD. However, for CBCL 88 (uncooperative), children with ASD responded to higher scores compared to children with DD. This suggested that, even though the subscales measure the same latent construct (as ODP here), depending on the condition of developmental disabilities, items might function accordingly with distinct symptoms.

Finally, one item in Affective Problems, i.e., CBCL 49 (overeating), which showed up as DIF, is somewhat surprising. Earlier studies showed that ASD children might tend to be picky on food selection [49] or may selectively overeat [50]. In some studies, children with DD showed overeating behaviors as well [51]. However, it was not clear about how children with ASD and those with DD differ on overeating behavior. From the response patterns of this item in our data (see Figure 1A), we can see that those children with DD and ASD are similar at lower levels of affective problems. However, once the level of trait goes higher, we found that children with DD scored much higher compared to children with ASD. This suggested that, at a high level of trait, children with DD have much more serious overeating problems compared to children with ASD, when they both have the same high level of affective problems.

### Limitation and Implication

There are several limitations of the current study. First, because previous studies on the factor structure of DSM-oriented scales of CBCL are scarce, and as there are also not many studies which compare the DD group and the ASD group, we proceeded with conservative moves with only full invariance models on our measurement invariance model analysis. This could expose us to the type-two errors that we possibly did not uncover the meaningful differences when we should. Further research is needed to replicate our findings. Secondly, our sample size is on the verge of sample size requirement for item response theory type of analysis (i.e., DIF we used here). However, all tested models converged, arguably because our sample size was insufficient. However, it is still possible that, with a much bigger sample, the results might be different. Last, in terms of generalizability, our sample was collected from South Asia. If we were compared to the western samples in earlier studies [16], our sample was quite different in terms of cultures or geographical locations. These differences limited the generalizability of our results. Yet, our findings delivered the perspective of cultural diversity. For diagnostic processes on children with developmental disabilities, this was emphasized on the latest revision of the DSM manual [52].

## 5. Conclusions

In terms of research on ASD, these two analyses, i.e., measurement invariance and DIF, were often separated with different publications in different papers [16,53]. However, previous studies exploring the measurement differences or similarities across populations showed that these two sets of analyses can be carried out together to discover the response differences. Researchers could further contrast these findings from the test level and the item level [54,55]. In addition, previous DIF studies were conducted with a comparison between people with ASD and typical population [15], but not children with DD. Therefore, we proceeded the study with a jointed approach to uncover the similarities and differences when two analyses were paralleled together. We hope to provide specific insights for clinicians on the use of CBCL DSM-oriented scales between the ASD group and the DD group. The findings showed that there were indeed some item differences. These differences manifested with behavioral patterns, even when the latent traits are at the same level, in which we would not find with typical analyses that focused on mean differences between items or tests. The clinical use of CBCL DSM has its benefit, but practitioners might want to pay attention to the latent individual differences on children with ASD or children with DD. Children with the same underlying latent traits of each subscale might mark different score patterns is the crucial take-away message here. 

## Figures and Tables

**Figure 1 children-09-00111-f001:**
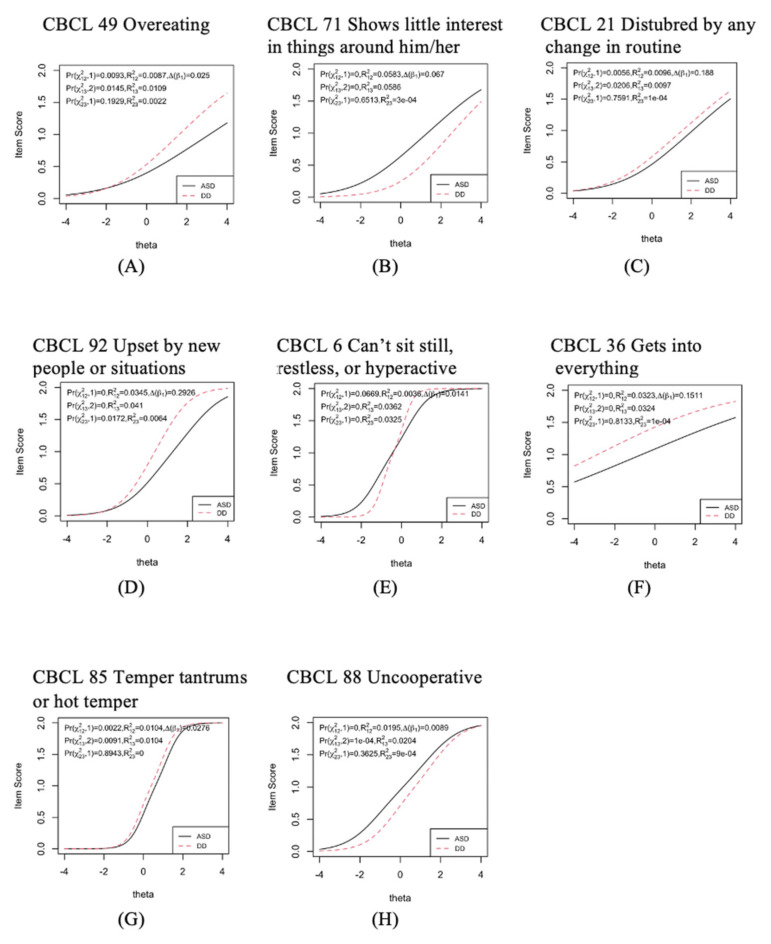
Comparison on DIF items’ score responses between groups across different trait levels. The solid line represents the ASD group. If one line is above another line in that area, this suggests that the above line score is higher. (**A**) The differences between ASD group and DD (higher) group appear to be at high affective problems level; (**B**)The differences between ASD (higher) group and DD group are across the spectrum of affective problems; (**C**) The small differences between ASD group and DD(higher) group appear to be at above average PDP level; (**D**) The differences between ASD group and DD (higher) group appear to be at high PDP level; (**E**) The differences between ASD group and DD (higher) group appear to be at both high and low attention deficit hyperactive problems level, but at high level, DD is higher, and at lower level, ASD is higher; (**F**) The differences between ASD group and DD (higher) group are across the spectrum of attention deficit hyperactive problems level; (**G**) The small differences between ASD group and DD(higher) group appear to be at above average oppositional defiant problems level; (**H**)The differences between ASD (higher) group and DD group are across the spectrum of oppositional defiant problems, but at lower level the difference is bigger.

**Table 1 children-09-00111-t001:** The group comparison of background variables.

	ASD (*n* = 228)	DD (*n* = 215)	*p*
CA (months)			
Mean (SD)	32.28 (9.16)	30.75 (10.10)	0.097
MA (months)			
Mean (SD)	21.02 (10.03)	23.94 (9.21)	0.002
Reporter			
Mother: Father	211:17	199:16	0.995
Parents’ years of education			
Mean (SD): mother	14.52 (2.39)	14.03 (2.55)	0.037
Mean (SD): father	14.56 (2.51)	13.65 (2.73)	<0.001
ADOS total scores ^a^			
Mean (SD): Module 1	17.19 (3.13)	3.38 (2.33)	<0.001
Mean (SD): Module 2	15.77 (3.09)	3.00 (1.86)	<0.001
Gender			
Male: female	203:25	148:67	<0.001

CA = chronological age; SD = Standard Deviation MA = mental age; ADOS = Autism Diagnostic Observation Schedule; ASD = autism spectrum disorder; DD = developmental delays. ^a^ 418 children (ASD:215, DD:203) were assessed with module 1 and 25 children (ASD:13, DD:12) were assessed with module 2.

**Table 2 children-09-00111-t002:** Model comparison of CBCL DSM-oriented subscales (WLSMV estimation).

	$\;X^{2}$ $\left(D\;X^{2}\right)$	df $\left(Ddf\right)$	$p\;\mathrm{Value}\;\mathrm{of}\;DX^{2}$ Test	RMSEA (Δ RMSEA)	SRMR (Δ SRMR)	CFI (Δ CFI)
Aff						
Configural	145.691	70		0.070	0.094	0.920
Configural vs. Metric *	147.711 (2.02)	79(9)	0.358	0.063 (−0.007)	0.099 (0.005)	0.927 (0.007)
Metric vs. Scalar	193.702 (45.991)	88(9)	<0.001	0.074 (0.011)	0.099 (0)	0.888 (0.039)
Anx						
Configural *	134.905	70		0.065	0.089	0.944
Configural vs. Metric	157.449 (22.544)	79(9)	0.019	0.067 (0.002)	0.101 (0.012)	0.932 (−0.012)
Metric vs. Scalar	NA	NA	NA	NA	NA	NA
PDP						
Configural	334.265	130		0.084	0.104	0.830
Configural vs. Metric *	320.441 (−13.824)	142(12)	0.48	0.075 (−0.009)	0.108 (0.004)	0.852 (0.022)
Metric vs. Scalar	368.433 (47.992)	154(12)	<0.001	0.079 (0.004)	0.108(0)	0.822 (−0.030)
ADHP						
Configural	190.997	18		0.209	0.129	0.871
Configural vs. Metric *	186.431 (−4.236)	23(5)	0.13	0.180 (−0.029)	0.132 (0.003)	0.878 (0.007)
Metric vs. Scalar	233.757 (47.326)	28(5)	<0.0001	0.183 (0.03)	0.130 (−0.002)	0.847 (−0.031)
ODP						
Configural	64.899	18		0.109	0.059	0.966
Configural vs. Metric *	59.243 (−5.656)	23(5)	0.38	0.085 (−0.024)	0.064 (0.005)	0.973 (0.007)
Metric vs. Scalar	98.809 (39.566)	28(5)	0.0001	0.107 (0.022)	0.062 (−0.002)	0.948 (−0.025)
It is non invariant if…				≥0.015	≥0.030	≥−0.010

Aff = Affective Problems; Anx = Anxious Problems; PDP = Pervasive Developmental Problems; ADHP = Attention Deficit/Hyperactivity Problems; ODP = Oppositional Defiant Problems. * showed the indication of the best fit, and numbers in () showed the differences. “NA” indicated that next comparison was not proceeded because the previous invariance was not achieved.

**Table 3 children-09-00111-t003:** DIF items on each subscale.

Subscale	Items
Affective Problems	CBCL 49 Overeating
	CBCL 71 Shows little interest in things around him/her
Pervasive Developmental Problems	CBCL 21 Disturbed by any change in routine
	CBCL 92 Upset by new people or situations
Attention Deficit/Hyperactive Problems	CBCL 6 Can’t sit still, restless, or hyperactive
	CBCL 36 Gets into everything
Oppositional Defiant Problems	CBCL 85 Temper tantrums or hot temper
	CBCL 88 Uncooperative

## Data Availability

Data sharing not applicable. No new data were created or analyzed in this study. Data sharing is not applicable to this article.

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
