# Peer review of "How Do Children with Autism Spectrum Disorder and Children with Developmental Delays Differ on the Child Behavior Checklist 1.5–5 DSM-Oriented Scales?"

_children, 2022, doi:10.3390/children9010111_

Round 1

Reviewer 1 Report

I consider the work to be very interesting for the field of neurodevelopmental disorders, specifically for ASD. The article could be published as is, although as a suggestion to the authors of the article, I believe that the sample of subjects with ASD could have been described further, specifically taking into consideration the degree and index of autism.

Author Response

Reviewer 1

I consider the work to be very interesting for the field of neurodevelopmental disorders, specifically for ASD. The article could be published as is, although as a suggestion to the authors of the article, I believe that the sample of subjects with ASD could have been described further, specifically taking into consideration the degree and index of autism.

Response to reviewer 1

Thank you so much for your very positive and helpful feedbacks! We appreciate the suggestion on the degree and index of autism – we have added a bit more description on this in the section. Please see page4, line147-150. 

‘and they exhibited a minimum of three deficits in social communication and social interaction and two restricted/repetitive behaviors. In addition, these children also belonged to autism or autism spectrum (i.e., pervasive developmental disorder-not otherwise specified, PDD-NOS) based on the ADOS classification.’ (p.4, line 147-150)

Reviewer 2 Report

Dear Authors,

Thank you for the opportunity to revise your work. This is a very interesting topic. Indeed, the differential item functioning analysis is an interesting tool for investigating the invariance of the measurement.

Please find below some comment that may increase the clarity of the manuscript.  

Lines 85- 96. You may add some example of the use of the differential item functioning analysis in questionnaire investigating disability and behavioral disorders, e.g., Geri T, et al. Rasch analysis of the Neck Bournemouth Questionnaire to measure disability related to chronic neck pain. J Rehabil Med. 2015 Oct 5;47(9):836-43. doi: 10.2340/16501977-2001,  Pellicciari L, et al., 'Less is more': validation with Rasch analysis of five short-forms for the Brain Injury Rehabilitation Trust Personality Questionnaires (BIRT-PQs). Brain Inj. 2020 Dec 5;34(13-14):1741-1755. doi: 10.1080/02699052.2020.1836402. 

Line 327. According to the results, “none of the scales achieved scalar invariance”. It is not clear why the authors did not adjust the factor structure by using the Modification Indexes.

Consider to add an appendix with the factor loadings for all the items of the Child Behavior Checklist 1.5–5, divided according to subscale.

Figure 1. A note to explain the figure should be added. Which of the group scored higher?

Author Response

Reviewer 2

Thank you for the opportunity to revise your work. This is a very interesting topic. Indeed, the differential item functioning analysis is an interesting tool for investigating the invariance of the measurement.

Response to reviewer 2

Thank you so much for your very kind and helpful comments and suggestions!

Please find below some comment that may increase the clarity of the manuscript.  

1.Lines 85- 96. You may add some example of the use of the differential item functioning analysis in questionnaire investigating disability and behavioral disorders, e.g., Geri T, et al. Rasch analysis of the Neck Bournemouth Questionnaire to measure disability related to chronic neck pain. J Rehabil Med. 2015 Oct 5;47(9):836-43. doi: 10.2340/16501977-2001,  Pellicciari L, et al., 'Less is more': validation with Rasch analysis of five short-forms for the Brain Injury Rehabilitation Trust Personality Questionnaires (BIRT-PQs). Brain Inj. 2020 Dec 5;34(13-14):1741-1755. doi: 10.1080/02699052.2020.1836402. 

Response to Reviewer 2. 1:

We very much appreciate that reviewer 2’s suggestions on adding a few examples here. In addition, following the reviewer’s suggestions we found another two literatures (i.e., Bruckner, et al., 2007.; Conrad et al., 2007). These articles have been added in line 92 (p.2)  & p.14.

‘particularly on disabilities [9,10,11,12]’(p.2, line 92)

2.Line 327. According to the results, “none of the scales achieved scalar invariance”. It is not clear why the authors did not adjust the factor structure by using the Modification Indexes.

Response to Reviewer 2. 2:

Thank you so much for your helpful comment! Although the choice of parameters to be relaxed can be generated from the suggestion of modification index in most softwares, such a choice is data driven and could, sometimes, potentially result in type 1 errors as well (please see Millsap & Kwok, 2004’s discussion, p.94). As there are very few papers that had compared between Autism and DD currently, we do not have clear clues about these choices. Therefore, we chose to proceed with the full invariance models only. Please see line 276-282 (p.6)

‘Partial invariance is an additional test to relax the problematic parameters and keep other parameters invariant to improve the model fits. Although the choice of parameters to be relaxed can be generated from the suggestion of modification index in most software, such a choice is data driven and it could, sometimes, potentially result in type 1 errors as well [44]. Because there are not many papers that had compared between these two groups of children currently, we are not sure about these choices of relaxing parameters. Therefore, we chose to proceed with the full invariance models only.’ (p.6, line 276-282)

3.Consider to add an appendix with the factor loadings for all the items of the Child Behavior Checklist 1.5–5, divided according to subscale.

Response to Reviewer 2. 3

Thank you so much for this suggestion. We have added this table as appendix. (p.17-18)

4.Figure 1. A note to explain the figure should be added. Which of the group scored higher?

Response to Reviewer 2. 4

Thank you so much for this suggestion as it helped us understand the figure was not clear. We have added a note as the explanation for the readers. The line 452-453 (p.11) is the group score higher.
